# Applications of Reciprocal Teaching in Flipped Classroom to Facilitate High Level of Cognition for Sustainable Learning Practices

**Wu-Yuin Hwang** [1,2] **, Tsu-Hsien Wu** [1] **and Rustam Shadiev** [3,*]

1 Graduate Institute of Network Learning Technology, National Central University, Taoyuan 32001, Taiwan
2 Department of Computer Science and Information Engineering, National Dong Hwa University, Hualien 974301, Taiwan
3 College of Education, Zhejiang University, Hangzhou 310058, China
* Correspondence: rustamsh@gmail.com

**Abstract:** In traditional engineering education, students usually have little training on project implementation. Therefore, students have few chances to cultivate and develop their high-level cognitive abilities for the sake of achieving sustainable learning practices. We carried out two consecutive studies to overcome this issue. In both studies, we incorporated a flipped classroom approach into the project-based engineering education curriculum. Twelve junior graduate students majoring in electrical engineering participated in Study 1, and ten junior graduate students participated in Study 2. They all took the Signal Processing of Power Quality Disturbances class and practiced their skills in a computer lab, using LabView software. After we found from the results of Study 1 that the learning activities placed a heavier load on students and their advanced cognitive skills were not developed well, the reciprocal teaching method was introduced to students in Study 2. We assumed that the reciprocal teaching method could improve these outcomes, as well as achieve sustainable learning practices. The results demonstrated that students' load in Study 2 was reduced, and their high-level cognitive skills improved compared to those in Study 1. Based on these results, we conclude that the reciprocal teaching method can be incorporated into the flipped classroom during project-based engineering education, as it helps prevent students from becoming overloaded, facilitates cognitive abilities from basic to high, and ensures sustainable learning practices.

**Keywords:** engineering curriculum; project-based learning; flipped classroom; reciprocal teaching

## 1. Introduction

At present, engineering courses such as the Signal Processing of Power Quality Disturbances course are mostly theoretical and focus on the derivation of relevant theories and models. Most of them aim to equip students with necessary knowledge, which is the basic level of cognition. Therefore, high cognitive abilities, such as application of new knowledge to solve real-life problems or creativity, are overlooked by the instructors. That is, most engineering classes are organized in a way in which the instructor delivers lectures and students passively acquire knowledge. In such circumstances, students have very limited chances to develop their higher-level cognition.

Furthermore, for some students—especially those with active experimenter and reflective observer learning styles—classes that emphasize lecture over practice are not very useful. Thus, there is a mismatch between the learning styles of engineering students and teaching styles of engineering instructors. As a consequence of this mismatch, students become bored and inattentive in class, perform poorly, have low learning motivation, and, in some cases, they change or even drop out of such courses. Instructors, confronted by low learning outcomes, such as test scores, unresponsive or hostile classes, poor attendance, and dropouts, know that something is not working. They may become overly critical

toward their students, thus making things even worse. As a result, society loses potentially excellent engineers [1]. From the above, we can understand that students in the field of engineering education lack experience in cooperation and expressing their ideas, as well as necessary practical skills. Furthermore, the research on engineering education rarely explores learners' cognitive development in practice, especially with respect to high cognitive levels, i.e., the application of newly learned knowledge to new contexts or creativity.

Such problems need to be addressed by educators and researchers. In addition, sustainable teaching and learning practices should be achieved. That is, such practices take place when educators and researchers equip their students with the skills and strategies that help them engage in lifelong independent learning though various experiential project-based learning tasks that require research, critical thinking, and collaboration.

In this research, we aimed to promote students' skills and enhance their high-level cognitive abilities by incorporating the reciprocal teaching method in the flipped classroom into a project-based engineering curriculum and achieve sustainable teaching and learning practices. This study aimed to address the following research questions: What are learning experiences and outcomes of the students studying the signal processing of power quality disturbances under the project-based learning approach and flipped classroom strategy? How can implementation of the reciprocal teaching method facilitate the learning experiences and outcomes of the students?

## 2. Literature Review

### 2.1. Flipped Classroom

The flipped classroom has become a popular teaching strategy recently [2–6]. Fulton [7] claimed that the flipped classroom is advantageous for learning because (1) students move at their own pace; (2) doing "homework" in class gives teachers better insight into student difficulties and learning styles; (3) teachers can more easily customize and update the curriculum and provide it to students 24/7; (4) classroom time can be used more effectively and creatively; (5) teachers using the method report seeing increased levels of student achievement, interest, and engagement; (6) learning theory supports the new approaches; and (7) the use of technology is flexible and appropriate for "21st century learning." For these reasons, the flipped approach was successfully applied in engineering education [3–6]. For example, Mavromihales and Holmes [3] presented a method to deliver a workshop based on the flipped learning approach. The scholars explored whether the flipped classroom approach can enhance the learning experience through better engagement with the students as compared to conventional classroom-based learning. The level of student participation and level of success were established in the study. Merrett [4] combined flipped classroom instruction, case-based learning in an active classroom, and authentic assessments in an Introduction to Engineering Materials Course. Merrett [4] found that flipped classroom instruction had a negligible effect on students' final exam performance compared to a traditional lecture mode, and case-based learning had a positive impact on students' quiz and laboratory scores. Therefore, Merrett [4] suggested the use of a flipped classroom approach with case-based learning in an active classroom, and authentic assessments are recommended for teaching engineering materials. Saterbak et al. [5] implemented and assessed a flipped classroom approach for first-year engineering design. The scholars implemented a flipped classroom approach that emphasized the development of higher cognitive levels for the students. Student learning was assessed, and outcomes from the flipped approach and the lecture approach showed no statistically significant differences because it was an inquiry-based course since its inception. Zhang and his colleagues [6] integrated mobile learning and SPOC-based flipped classroom to teach an engineering course.

Their approach included pre-class (knowledge acquisition), in-class (knowledge internalization), and after-class (knowledge application) stages. Zhang and his colleagues adopted the WeChat applet Mu classroom with m-learning technology in the in-class stage. They also designed several interactive activities based on the Mu classroom to improve the

teacher–student interactions. The results showed that, after using such an approach, the average score of the final exam improved, and the failed percentage decreased. Furthermore, positive feedback from students was received, stating that the approach was effective and motivated students' learning interests and knowledge understanding.

Finally, several pitfalls of the flipped approach were also reported in the literature. For example, students new to the method may be initially resistant because it requires that they do work at home rather than be first exposed to the subject matter in school. Consequently, they may come unprepared to class to participate in the active learning phase of the course [8].

### 2.2. Project-Based Learning

Project-based learning (PBL) is a model that organizes learning around projects [9,10]. Thomas [11] defines PBL as an approach which includes authentic content, authentic assessment, and student-centered learning activities with clear and detailed teaching goals. During the PBL learning process, learners must learn to find the problem out and have the ability to implement, collect, and integrate information and to train the communication skills with others through the group discussions, and try to propose a solution to the problem with others [12,13]. In order to promote the interaction between the teacher and students and students' ability to actively think about a problem and solve it in PBL flipping classrooms, the other most important thing is to inspire students to develop their higher level of cognition.

Many scholars explored how PBL approach can lead to a higher level of cognition, particularly in engineering education. For example, Nurbekova et al. [14] used the PBL approach in engineering education to teach mobile application development. The scholars explored the impact of the used approach on the students' cognitive skills. The impact of the approach was evaluated through the questionnaire, and its effectiveness was confirmed by the empirical data. Sulisworo [15] attempted to improve higher-order thinking skills through the project-based learning approach on STEM education settings. The experimental group learned under the PBL approach, and the control group was exposed to scientific learning. The results demonstrated the positive impact of the PBL approach on the students' higher-order-thinking skills.

### 2.3. Reciprocal Teaching

Reciprocal teaching routines force students to respond, even if the level of which they are capable is not yet that of an expert. However, because the students do respond, the teacher has an opportunity to gauge their competence and provide appropriate feedback. In this way, the procedure provides an opportunity for the students continuously make progress until they approach full competence [16].

Reciprocal teaching is an instructional procedure designed to teach students cognitive strategies that might lead to their improved comprehension [17]. Learning about cognitive strategies such as summarization, question generation, clarification, and prediction can be supported through dialogue between the teacher and students as they attempt to gain meaning from the learning content. Reciprocal teaching has two major features. The first is the instruction and practice of four comprehension-fostering strategies: question generation, summarization, prediction, and clarification. The second is the usage of reciprocal teaching dialogue as a vehicle for learning and practicing these four strategies. In reciprocal teaching, however, much greater emphasis is placed on encouraging students to provide instructional support for each other [18].

Reciprocal teaching received considerable attention in the field. For example, Zewail-Foote and Gonzalez [17] designed a crisscrossing learning experience (CCLE) course to promote a high level of collaboration, sense of ownership, and science identity among first-year students through the learning via the teaching paradigm and close mentoring. Nnamani et al. [19] researched the effects of reciprocal peer tutoring strategies on computer students' achievement. According to the scholars, the reciprocal peer tutoring strategy

had a significant effect on computer students' achievement in expository essay writing. Nnamani et al. [19] argued that expository essay writing skills are very important for computer students and that reciprocal peer tutoring should be adopted as a teaching strategy for expository essays in technical institutions. Reciprocal teaching strategies were applied in Shadiev et al. [20] to computer-programming learning. The scholars investigated the effects of reciprocal teaching strategies on learning outcomes. The results showed that the students who used reciprocal teaching strategies outperformed students who did not use them in regard to the level of cognition of program concept and program writing. The reason is that the reciprocal teaching strategies facilitate students to write program codes, as well as to explain them to their peers.

Informed by related studies, we applied reciprocal teaching in the flipped classroom to facilitate a high level of cognition for sustainable learning practices. The learning activity was designed by following the project-based learning (PBL) methodology. With such an integrative approach, we aimed to cultivate and develop engineering students' high-level cognitive abilities. As can be seen from the literature review, not many studies had such an integrative approach or focused on the development of engineering students' high level of cognitive abilities.

## 3. Methods

The research method was a case study [21]. According to Creswell [22], a case study is an in-depth exploration of a bounded system (e.g., activity, event, or process) based on extensive data collection. That is, in a case study, researchers focus on a program, event, or activity involving individuals or a group. We particularly employed a multiple instrumental case study. We focused on illuminating a specific issue (how to facilitate a high level of cognition for sustainable learning practices), with cases (Study 1 and Study 2) used to illustrate the issue. We described and compared cases to provide insight into an issue [8].

### 3.1. Participants and Research Architecture

The research architecture is shown in Figure 1. Our research was divided into two studies, i.e., Study 1 and Study 2. In Study 1, twelve junior graduate students majoring in electrical engineering participated. All of them were males. They were divided into six pairs.

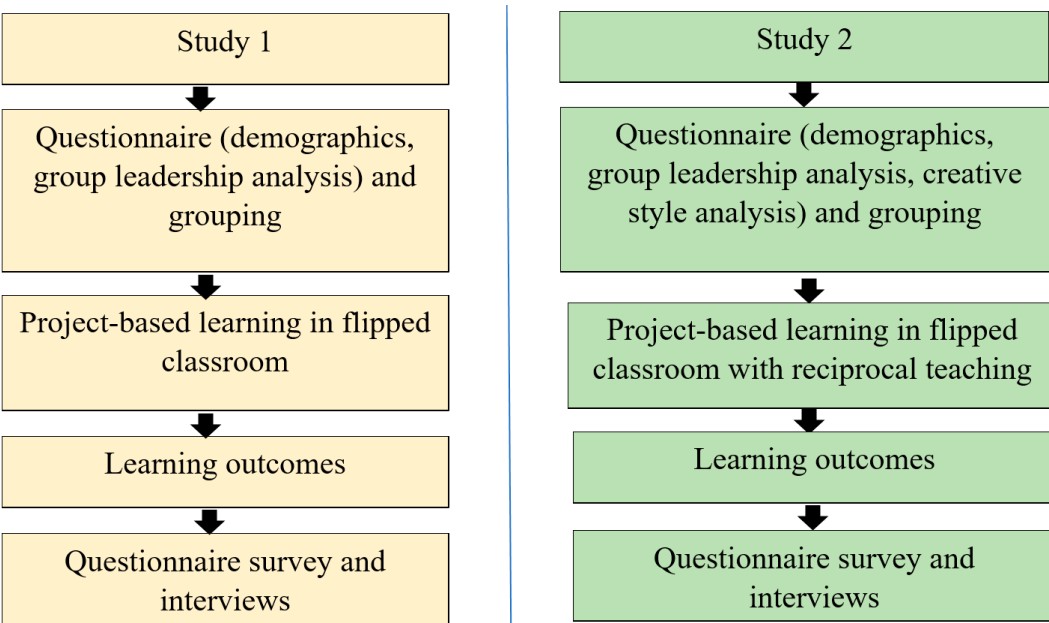

**Figure 1.** Research architecture.

The course administered in this study was Signal Processing of Power Quality Disturbances. The learning activity was designed following the project-based learning (PBL) approach in Study 1. Students worked on assigned real-world problems individually and collaboratively to obtain hands-on experience. A flipped classroom, an instructional strategy was used; that is, the students completed pre-class work, e.g., studying learning content and searching for certain information on the Internet to gain basic knowledge. Then, in class, they applied and mastered newly learned knowledge through problem-solving activities, discussions, giving/receiving feedback, reflections, and collaboration. The study was carried out in the computer laboratory and the participants used LabView (Laboratory Virtual Instrument Engineering Workbench) to design and simulate their ideas about power signal processing. LabView is a simulation software for testing, measuring, and controlling engineering design in the laboratories. It can help students to understand the specific knowledge of engineering applications through graphical programming and representation. Using displaying logic diagrams and algorithm analysis, the students can improve their practical experience and understanding of power signal processing. The students designed their ideas into products and then operated and debugged their products. The teacher and teaching assistant helped the students when necessary. For example, the students asked a teacher or teaching assistant questions when they encountered any difficulties. Therefore, after practical experience, the students could develop their high-level cognition and reach such cognitive levels as application or analysis (see Bloom's Taxonomy in Anderson et al. [23]). More details about LabVIEW software can be found from LabVIEW [24].

Every week, before the class, the students were assigned to develop a weekly preview of related learning content. The students had to summarize their ideas generated from their weekly preview, write them down, and then hand them in as a pre-class assignment report. Then, in class, each group was asked to do a presentation and share their ideas. Other groups had to give feedback or share their ideas related to the presentations. Finally, after class, each group was asked to revise its weekly preview report and hand in a revised version (called as a weekly report) by integrating peer feedback and ideas shared by others. Regarding monthly reports (or project reports), each group needed to hand in a project report monthly by integrating what the group members learned each month.

After investigating students' learning perceptions of their learning experiences through interviews, we found that the students required more time to practice their skills with hands-on tools, as this can enhance their understanding and application of knowledge learned in class to solve real-life problems. Moreover, we found that the students encountered difficulties in pre-class study and answering pre-class questions given by the teacher, as they often had a superficial understanding of learning content before class. Last but not least, students' higher level of cognition was not developed very well. Therefore, Study 2 was carried out after Study 1, with several modifications in the initial design of the learning activity.

In Study 2, ten junior graduate students with a major in electrical engineering participated. All of them were males. In order to motivate students to have more engagement and creativity in group discussion and collaboration (e.g., students with high leadership and creativity in each pair can lead discussions and brainstorm ideas to improve their pair learning), students were assigned to different pairs based on their creativity style and the personality related to leadership. To this end, the participants were surveyed using Big Five personality questionnaires and the creativity styles questionnaire [25]. Two dimensions of the Big Five personality, openness to experience and extraversion, were used to represent the potential of students' leadership. Finally, the normal S-type grouping based on scores of leadership and creativity was employed to divide students evenly into different groups. The students were divided into five pairs, and each pair had students with heterogeneous attributes in leadership and creativity. The learning activity followed the PBL approach, and the flipped classroom strategy was used. The same teacher instructed the participants in Study 1 and Study 2. Both studies were 18 weeks long, and the students learned about the electric circuit in the Signal Processing of Power Quality Disturbances course. The

only difference between the two studies was the reciprocal teaching approach that was introduced in Study 2. Reciprocal teaching, in this study, refers to student interaction with peers through explaining, questioning, and clarifying difficulties, new concepts, and applied methods that were used [20]. Reciprocal teaching included the following major dimensions (Figure 2). (1) Summary: Identify the main point about the content and make it become 2~3 sentences to capture what you read. (2) Ask questions: Listen to the problems in your mind, e.g., what will...? or how come...? and write it down. (3) Clarify: List all unfamiliar things, link with your prior knowledge, and then answer any questions. (4) Forecast: Expect those parts which will happen or connect next and write down your forecast.

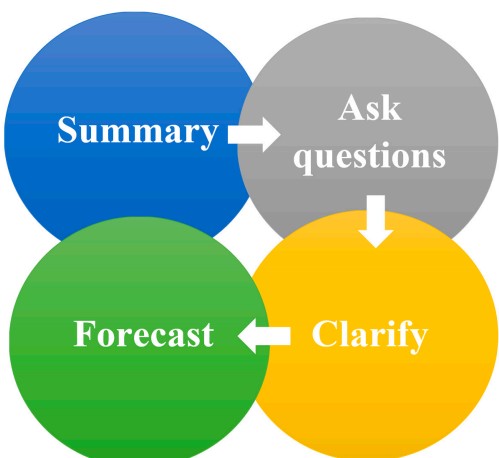

**Figure 2.** Reciprocal teaching activity.

Both collaborative learning (Study 1) and reciprocal teaching (Study 2) align with social constructivism theory [26]. According to this theory, learners are active participants in the creation of their own knowledge. It was suggested that a central notion in social constructivism is assisted learning, a concept that is influenced by socio-culturalism and its concept of proximal learning. Although both activities align with social constructivism theory, they are quite different in nature and, as we assumed, bring about different learning effects.

### 3.2. Research Instruments

The following research instruments were used: group leadership, creative style, project score, a post-test questionnaire, and interviews.

The group leadership questionnaire was developed based on the Big Five personality traits examination tool [27]. It measures the following learner characteristics: (a) openness to change, innovation, new experience, and learning; (b) conscientiousness, self-discipline, dutiful act, and aim for achievement; (c) extraversion—tendency to be sociable, talkative, and have positive emotions; (d) agreeableness—to be kind, sympathetic, cooperative, and considerate; and (e) neuroticism—experience unpleasant emotions (e.g., anger, anxiety, and depression) easily. The group-leadership data were collected in both studies.

The creative style questionnaire was adopted from the study by Kumar et al. [25]. There were 78 items in the questionnaire. Both the group leadership and creative style questionnaires were validated through scrutiny of the instructor and two experts. The creative style data were collected in Study 2 only.

Learning outcomes, such as scores for the weekly preview and report, as well as monthly report, were measured. Weekly previews and reports were measured right after they were submitted, and final project reports were evaluated at the end of the semester. We used the revised version of Bloom's Taxonomy [23] for the data analysis. Figure 3 shows different cognitive levels and the colors corresponding to them. Representing cognitive levels in corresponding colors was helpful in coding content created by the students.

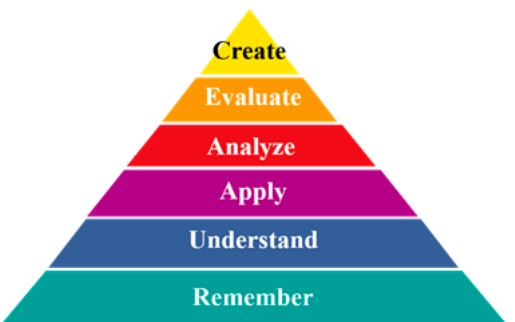

**Figure 3.** Bloom's Taxonomy.

Bloom's Taxonomy was also used. It includes six levels. The remember level refers to recognizing or recalling knowledge from memory. Remembering is when memory is used to produce or retrieve definitions, facts, or lists, or to recite previously learned information. The understand level represents constructing meaning from different types of functions, be they written or graphic messages or activities. The apply level consists of carrying out or using a procedure by executing or implementing it. The analyze level involves breaking materials or concepts into parts, determining how the parts relate to one another or how they interrelate, or determining how the parts relate to an overall structure or purpose. The evaluate level represents making judgments based on criteria and standards through checking and critiquing. Critiques, recommendations, and reports are some of the products that can be created to demonstrate the processes of evaluation.

We adjusted the definitions of the taxonomy as follows. Remember: Basic judgments for various power quality events (e.g., harmonics, flicker, voltage swell, voltage dip, etc.). Understand: Learn about the various algorithms' analysis methods, principles, and applicable power quality events. Apply: Use LabView software to simulate power events and implement the algorithm. Analyze: Students can analyze the algorithms of power events and understand how they work. Evaluate: Students comment on the merits and demerits of experiments based on the algorithmic norms and standards they learned and experiences they had. Create: Improve the original algorithm or propose innovative ideas and incorporate them into the group project at the end of the term.

We encoded student weekly previews and reports and monthly project reports. We used the color encoding of content based on Bloom's taxonomy. The data and content presented by the students were classified into six levels of the taxonomy, with different colors. For example, the green color represents the remember level, and the yellow color represents the create level (see Figure 3 for colors and related levels of the taxonomy).

After the encoding process, the statistics on the number of the six cognitive levels were performed. Finally, we explored and observed the relationship between trend changes and research variables.

A post-test questionnaire with several items was carried out at the end of each study. The first part of the questionnaire focused on whether students liked the after-school assignments. Everyone could preview them and cooperate with the group members to prepare a weekly report. It also focused on whether students felt that the peer interaction and learning quality improved. The second part focused on whether students felt that weekly previews and reports helped them better clarify and understand concepts of the course and improve their cognitive levels and abilities to solve problems.

The third part focused on students' views of the hands-on practice of LabView, i.e., whether it helped them learn content of the course and increase their practical experience. The fourth part focused on the mode of the project report, i.e., whether it helped them improve their problem-solving skills and solve problems. The fifth part focused on whether students felt that the homework of LabView can help them think and develop higher cognitive levels and whether they have learned different solutions and generated more ideas from the report. In the sixth part, we also explored whether students were still willing

to participate in similar teaching courses. The last item was an open-ended question, and the students were asked to write down any experienced issues associated with our instructional approach. The questionnaire scores of the first six parts were calculated by using a five-point Likert scale.

Interviews with students were also carried out. In the interviews, the students were asked about their learning experiences. The interviews' data were recorded, transcribed, and coded following the grounded theory design [21].

## 4. Results and Discussion

### 4.1. Study 1

Although we incorporated the flipped classroom and project-based learning approaches, our results showed that the students were not familiar with the simulation tools and lacked PBL experience. In addition, a large number of algorithmic theories caused students to spend more time on understanding the course content, and the results of the analysis showed that their six cognitive levels were more distributed in understanding, application of software, and analysis of data cognitive levels. There was not enough time for students to have more growth at other higher cognitive levels and to design and produce more innovative projects.

#### 4.1.1. Weekly Preview and Report

After six weekly previews and reports, we analyzed the distribution trend of each cognitive level in sequence (see Figure 4). According to the figure, students' cognitive level was distributed more in the understand and analyze levels. That is, students were more focused on understanding theoretical concepts by finding related information and then discussing and analyzing it with other students.

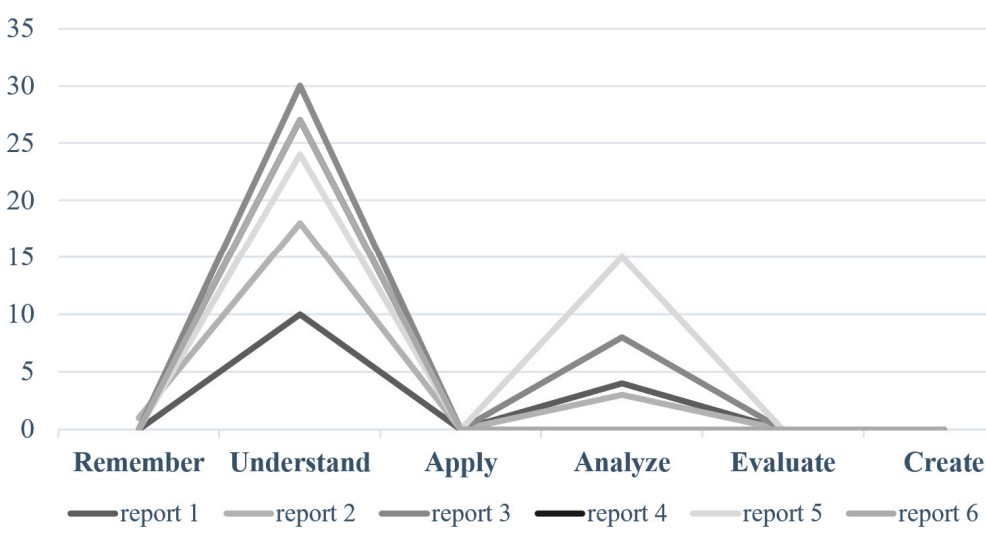

**Figure 4.** The trends in different cognitive levels at six stages.

#### 4.1.2. Weekly Homework

Figure 5 shows the results of the cognitive-level evaluation in regard to homework. According to the results, the trend of cognitive level distribution for the understand, apply, and analyze levels was declining. The reason was because the theory of the course content was becoming more difficult. Students mentioned several reasons to explain this finding (see Appendix A).

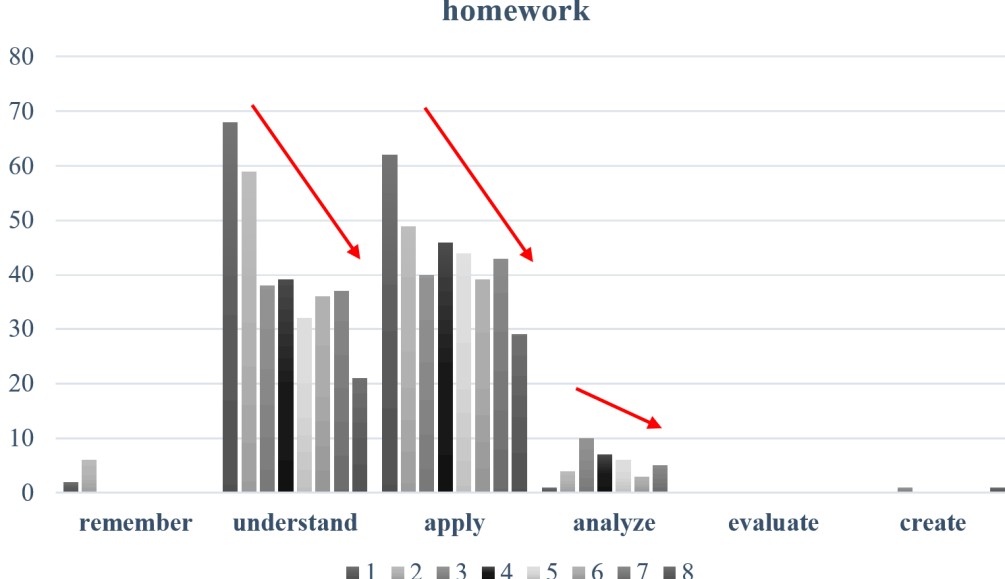

**Figure 5.** Gain value change for each cognitive level in homework (red arrows demonstrate declining trend).

In addition, we found that there was no significant change between the evaluate and create levels of cognition. Therefore, this part of the phenomenon and corresponding reason was one of the focuses of our Study 2 on the adjustment strategy.

4.1.3. Monthly Project Report

The trend of cognitive level change in each reporting stage is shown in Figure 6. From these data, we can observe that the cognitive level related to the understand, apply, and analyze levels improved during the course. However, there was no change in such cognitive levels as evaluate and create. Because these are the highest cognitive levels and very important for learning, in Study 2, we focused on enhancing them.

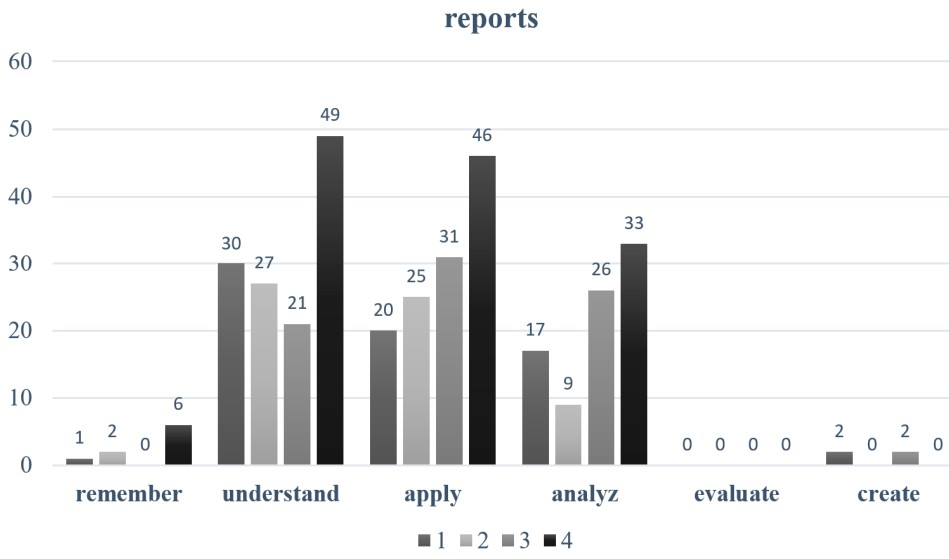

**Figure 6.** Gain value change for each cognitive level in four project reports.

### 4.1.4. A Post-Test Questionnaire from Study 1

The score of the questionnaire was 3.97 points, indicating that the students agreed that the course was helpful and that they liked it. However, student answers to an open-ended question showed that there was still considerable space for improvement. Some extracts from interviews with students are reported in Appendix A.

### 4.1.5. Other Findings

Weekly homework and monthly project reports require a lot of time to prepare. Monthly project reports affect students' performance more compared to weekly homework; however, students have to spend more time on monthly project reports than on weekly homework.

### 4.2. Study 2

Based on the results from Study 1, we modified our instructional approach by incorporating reciprocal teaching activities in Study 2. We grouped the students based on their leadership and creative styles in order to improve their abilities and facilitate their high cognitive levels. For example, students can learn new knowledge and then apply it to solve real-life problems.

### 4.2.1. Weekly Preview and Report

The instructor guided students to explore assigned problems. In addition, the instructor helped students understand the learning content and key points of the course. The interview results showed that this kind of teaching mode could promote students' active thinking and self-learning and also enhanced learning motivation, learning attitude, and communication ability. We provide some excerpts from the interviews in Appendix A.

Figure 7 shows the number of cognitive levels in each weekly preview and report. According to the figure, the understand part of the cognitive level greatly improved over time. In addition, the analyze and evaluate levels, which are higher cognitive levels, emerged.

**Figure 7.** The number of cognitive levels in each weekly preview and report.

Figure 8 presents trends in different cognitive levels. According to the figure, fluctuations in cognitive levels evolve from low cognitive levels (on the left side) to high cognitive levels (on the right side). The experiment was not long enough (i.e., only 18 weeks), and so time was not sufficient for the students to improve their professional knowledge and ability.

We suggest that students need long-term exposure and that the educators and researchers can perform similar studies in the future which could last at least one year. As a result, the change trend of cognitive levels can obviously advance, i.e., higher frequency of high cognitive levels, such as evaluate and create.

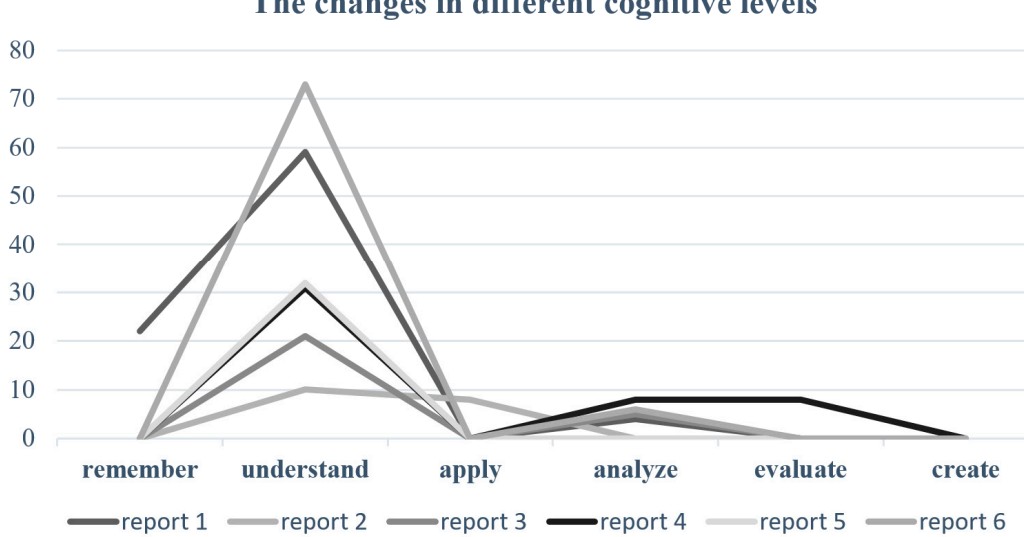

**Figure 8.** The changes in different cognitive levels.

4.2.2. Reciprocal Teaching Activity

In interviews, the students mentioned that they felt that, through this activity, they could train their teamwork, expression ability, leadership, understanding of content, and active thinking. In this way, their understanding of professional knowledge became more thorough and substantial, and the students could quickly review the content of the course again and deepen their memory. Furthermore, after discussing with each other, the students could also learn the methods that they have never thought of from different perspectives. Student opinions from interview are provided in Appendix A.

Figure 9 shows the frequency of different cognitive levels in each activity. According to the figure, as the course progresses, changes in the understand level's frequency from the first to the fifth activities decreased; however, it is the highest in the last activity. In the following, we provide some possible reasons. First, in the learning process, the algorithm architecture is a difficult and complex concept, so the interpretation, examples, discrimination, narrative, and interpretation that students can put forward on the theme of the algorithm are gradually reduced. Therefore, the change in the understand level throughout the activities declined. Second, the understand level in the last activity is the highest because of the subject matter of the discussion. That is, the students learned about algorithms related to signal processing of power quality disturbances in the first five activities; however, they judged power events in the last one, so it was easier for the students to identify, describe, and explain related concepts.

After comparing Figure 7 with Figure 9, it is observed that, in the learning process, there is a negative correlation between the understand level in the two figures, and the trend changes between "weekly preview and report" and "reciprocal teaching activity" are reversed (i.e., from the second activity to the fifth one). The important reason to explain this finding is that in terms of the "weekly preview and report", the students had more time to work and prepare it, but for the "reciprocal teaching activity", the students had less time to discuss and think about related concepts. Furthermore, although the frequency of the understand level in reciprocal teaching activities is less than in weekly preview and reports, "reciprocal teaching activity" can complement "weekly preview and report", thus, allowing learners to explore more different dimensions that were not found in the weekly preview and reports. The frequency of the understand level in "reciprocal teaching activities" may

be because the contents of the gain values were not included in the "weekly preview and reports", as shown in the figure.

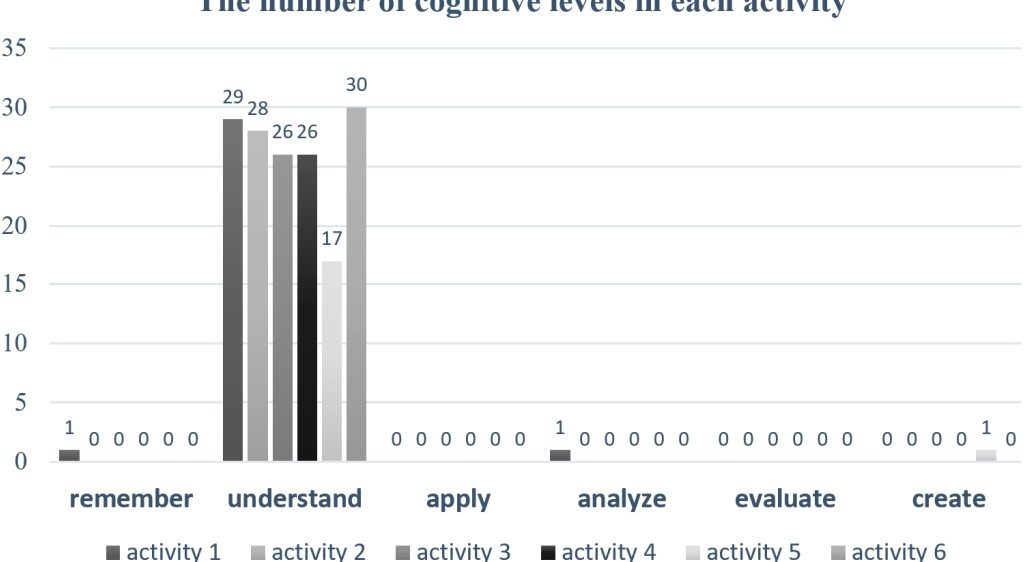

**Figure 9.** The number of cognitive levels in each activity.

### 4.2.3. Hands-On Practice Using LabView

In engineering education, hands-on practice plays a very important role. Hands-on practice not only extends and deepens the learning experience but also cultivates the ability to think critically and solve problems. More importantly, hands-on practice improves the cognitive level of learners. According to the curriculum, students are required to practice on LabView. If there is any problem in the process, students can raise their hands to ask for help from the teacher. Student opinions are provided in Appendix A.

Students have sufficient time in three classes to perform and operate the LabView. The teacher was there to assist them. Hands-on practices of LabView helped students validate and consolidate the knowledge concepts of the course. After practical experience, students were also able to reach such a high cognitive level as the apply and analyze levels.

### 4.2.4. Homework

Although the teacher's guidance and peer-to-peer interaction in the classroom can be used to stimulate the students' higher cognitive level, a low cognitive level was still seen in the majority. This suggests that these teaching strategies and the course design in the classroom were not enough to facilitate higher cognitive levels in engineering education. Therefore, we added another hands-on practice course of LabView and the practical assignments to the students, so that the students could practice after class. There were five assignments in total. The first four were the implementation of the algorithm from the simple to the hard, and the fifth one was to judge and implement the power quality event.

According to Figure 10, it can be found that the statistical chart is different from that of the weekly preview and reports and reciprocal teaching activities. In addition to the understand level, other high cognitive levels, i.e., apply and analyze, showed significant growth. However, the growth of the create level is not obvious, but it also appears as a sprout. Our results suggest that high cognitive levels need to be developed step by step through the process of training, brewing, fermentation, and then maturity. Although we added hands-on practice in the classroom and practical homework to cultivate students' ability, the teaching time (18 weeks) was not sufficient enough. In addition, it is necessary to administer diverse teaching activities and content to help students develop their higher cognitive abilities.

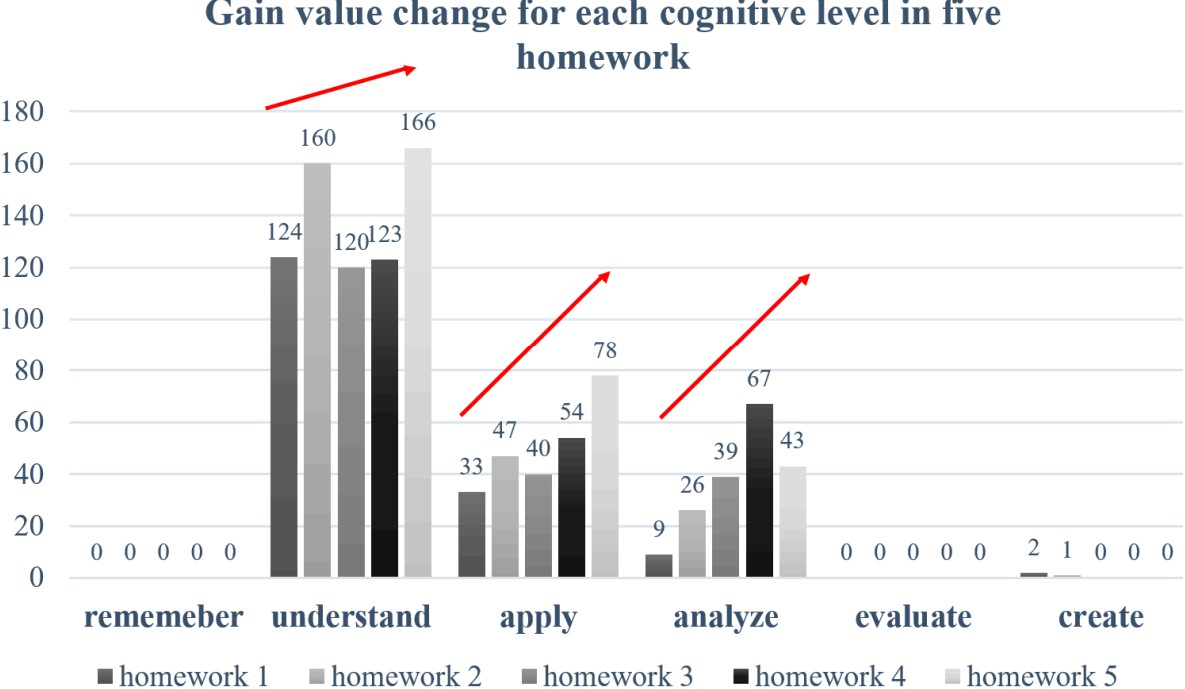

**Figure 10.** Gain value change for each cognitive level in five homework assignments (red arrows demonstrate rising trend).

### 4.2.5. Monthly Project Report

It can be found from Figure 11 that the gain value of the cognitive levels related to the second report is lower than that of others. This result may be due to the longer preparation time and more sufficient content and information in the first report. In addition, the report content was also reduced because the time interval of the second report with the first report was only three weeks, which was not enough. Another reason may be that the algorithm of the second report was more difficult, and the theory was constructed in the previous report. The students needed to spend more time on learning, understanding, implementing, verifying, and preparing reports so that the changes of the gain value at all cognitive levels were lessened in a short preparation time. In the third report, the analyze cognitive level was greatly improved because, after the implementation and application, the students needed to analyze, judge, and describe the information obtained. In addition, we could find that, through the project report method, the students' cognitive abilities not only remained in the remember or understand levels (see Figure 11) but also reached higher levels, such as apply and analyze.

According to the questionnaire and interview survey, more than 70% of the students agreed that they were active and willing to explain the results and share experiences with others in the project report. Student opinions are provided in Appendix A.

The instructional approach was student-centered, the learners discussed the projects with each other, and the teacher guided them at the right time when they became confused about some concepts. Such an approach enhanced the development of higher cognitive levels of the students, helping them achieve sustainable learning practices when compared to the traditional teaching approach.

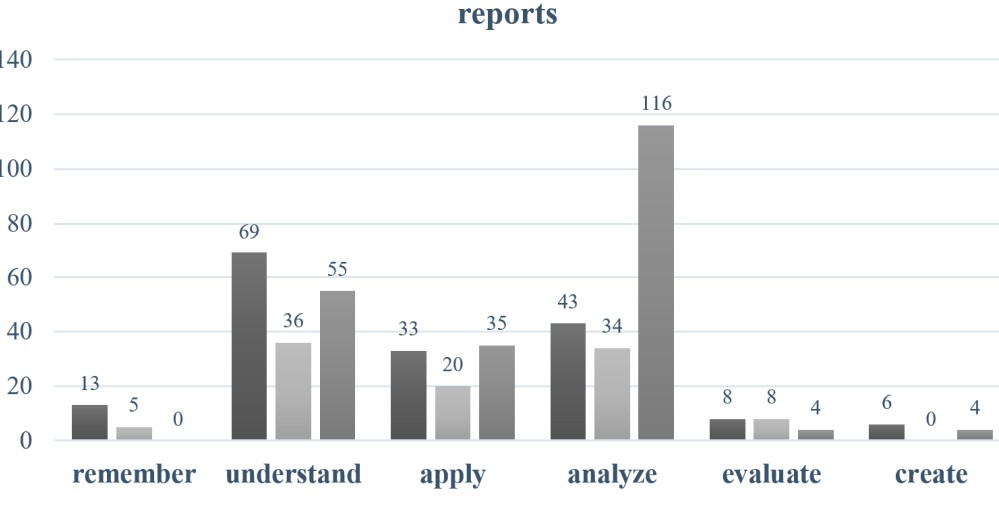

**Figure 11.** Gain value change for each cognitive level in three project reports.

4.2.6. Interview Results

From the interviews, we aimed to gain a deeper understanding of the students' experiences and obtain their suggestions. In the simulation of LabView, the students believed that, through implementation, they could be clearer about the application. When they achieved the goals, they could also have a great accomplishment and enhance the ability to express and work with the team. However, there were a few cross-disciplinary students who had also taken this course. Because LabView was not used normally for them, they believed that they were more passive in learning, and their achievement and satisfaction were relatively low. However, with the application of LabView, most of the students believed that they could learn the abnormal conditions of various power systems, the types of power quality events, and how to use different methods to do the test. They also had a better concept of and improvement on signal analysis. The students no longer just engaged in empty talk.

In the reciprocal teaching activities, the students could stimulate the thinking ability of each other through joint problem-solving. Some students also indicated that, because of their lack of knowledge in this field, it was relatively difficult to lead the group to discuss and analyze the problem. In this course mode, the students could stimulate their high-level thinking; develop their understanding of the application, details, and problems of the implementation; improve on the lack of theory; and increase their own ideas.

In the heterogeneous grouping of leadership and creativity styles, in addition to increasing the students' sense of responsibility for the supervision of the group's progress, the leader's influence on the learning among the members of the group was not obvious. From the students' responses, it could be found that the professional ability and the prior knowledge of the members occasionally revealed the gaps between each other; therefore, all contribution of the group often fell on the same students. How to make the students in the group help improve each other's abilities becomes an issue worth investigating. Finally, the students felt that the teacher could have the interactions and communicate with them during the free time of teaching, making them feel more likely to learn the content of the class and feel fulfilled.

### 4.3. Comparison between Study 1 and Study 2

We compared the results from Study 1 with those obtained in Study 2 (Table 1). The create level was the highest cognitive level, as it could help students develop their abilities on the creative dimension. However, as teachers, we need to be aware of the level of cognition of each activity in the course arrangement so that we can ensure that students are given a diverse curriculum experience and help them to promote higher-level thinking.

There is one thing which is very important and must not be ignored—if students do not have the lower level of cognitive skills, such as remember and understand, then they are not able to remember and understand what they have learned in the course. In that kind of situation, they are unlikely to apply what they have learned to the new environment, and there is no need to further analyze, evaluate, and develop products and generate new ideas through learning.

**Table 1.** Comparison between Study 1 and Study 2.

| Items | Study 1 | Study 2 |
|---|---|---|
| Group mode | Free grouping | Heterogeneous grouping (leadership and creative style) |
| Weekly preview and reports | Cognitive level represented understand and analyze levels; There was lack of interaction time with teachers. | Adding reciprocal teaching activities lead to gradual development of higher cognitive levels; The students had more time for interaction with teachers. |
| Homework | Once a week; The students felt burdened and time-consuming; The cognitive level represented understand and apply; Cognitive level frequency decreased over time. | Once in two weeks; Through flexible adjustment, the students were able to coordinate activities easier and further study; The cognitive level reached understand, apply, and analyze; Cognitive level frequency increased over time. |
| Monthly project reports | Four times; The schedule was more compact; The content repetition rate was high; Higher cognitive level frequency had no obvious change. | Three times; The schedule was more flexible; Hands-on practice; Higher cognitive level frequency had obvious change. |
| Advantages and disadvantages | The students cannot practice in class; Compact course activities and progress; The students had no improvement in high cognitive levels. | The hands-on practice activity was introduced; Flexible curriculum activities and schedule design; The students had improvement in high cognitive levels. |

As the old saying goes, "The nine-story platform starts from the soil" and "the high-rise building starts from the ground." Therefore, we use the project-based learning approach as the core of the two studies, and we also introduce the flipped classroom and the reciprocal teaching activities to the students to increase their mastery of the course content. In addition, the hands-on practice was implemented in order to develop student application and analytical skills. Meanwhile, we guided the students to reach high cognitive levels in this study.

In Study 2, the students had more interaction time in reciprocal teaching activities, although the teacher's teaching time was reduced, but relatively, there were more opportunities for the teacher to observe the student interaction. The teacher could approach different groups to observe whether there was good discussion, and then the teacher could also join the students in a timely manner to help them solve the problems and correct their mistakes. The teacher could also adjust the pace of teaching so that the teacher and the students had good interactions.

In Study 2, the weekly preview and reports still brought an additional course burden. Through the students' responses in the classroom, the teacher adjusted the teaching progress and content in time to meet the students' learning pace. Moreover, the curriculum load was reduced.

In addition, the hands-on practice of Study 1 was mostly executed after class, so it was impossible to strike while the iron was hot. We found student feedback in Study 1 showing that the students hoped to have direct practice drills in the classroom. After that, we observed the difference between Study 1 and Study 2 in the classroom, and we found that the hands-on practice time of the classroom of Study 2 was more about half than that of Study 1. This improved the students' application ability. In Study 2, the teacher also asked the assistant to teach the students LabView software so that students became familiar with the tool faster and achieved their learning goal faster.

The reciprocal teaching activities had more advanced or professional discussion, which improved the students' higher cognitive levels. Although the teacher's teaching time was shortened, the teacher's role and position were changed to that of expert and consultant. Additionally, at the part of the homework, the students increased the experience and ability

of practical operations by doing, so that students no longer thought about how to complete the project and do the surface work.

4.3.1. Weekly Preview and Report

Study 1 and Study 2 both contained six weekly previews and reports. We encoded each weekly preview and report and then compared. An independent t-test was used to explore the differences between Study 1 and Study 2. Results are reported in Table 2. It was found that there was no significant difference between Study 1 and Study 2 in each cognitive level. The reason for such a finding can be discerned from Figures 4 and 7. In the two studies, the most frequent cognitive level is the understand level in weekly preview and reports because the teacher provided a new question right before weekly preview and reports. The learning content was new for the students, so that they spent a lot of time exploring and understanding it. The frequency of other levels was low because higher cognitive levels could not be improved until the students understood new learning content well. Therefore, there is no significant difference at each cognitive level in the weekly previews and reports of two studies.

**Table 2.** Cognitive levels assessment of weekly preview and reports of Study 1 and Study 2 and their comparison.

| Cognitive Levels | Group | Mean | SD | *t*-Value | *p*-Value |
|---|---|---|---|---|---|
| Remember | Study 1 | 0.17 | 0.408 | −0.954 | 0.384 |
| | Study 2 | 3.67 | 8.981 | | |
| Understand | Study 1 | 22.67 | 7.421 | −1.476 | 0.191 |
| | Study 2 | 37.67 | 23.763 | | |
| Apply | Study 1 | 0 | 0 | −1.000 | 0.363 |
| | Study 2 | 1.33 | 3.266 | | |
| Analyze | Study 1 | 5 | 5.727 | 0.434 | 0.674 |
| | Study 2 | 3.83 | 3.251 | | |
| Evaluate | Study 1 | 0 | 0 | −1.000 | 0.363 |
| | Study 2 | 1.33 | 3.266 | | |
| Create | Study 1 | 0 | 0 | | |
| | Study 2 | 0 | 0 | | |

However, when we compare two figures (i.e., Figures 4 and 7), we can find that the development of the remember, understand, apply, and evaluate levels in Study 2 are better compared to those in Study 1. This is due to adjustments in teaching strategies and new curriculum arrangements which gradually stimulated students' development in other cognitive abilities.

In order to improve higher cognitive levels in weekly previews and reports, the instructors may consider changing the nature of the assigned weekly preview and reports [28–31]. Initially, the students were asked to preview learning content before class and summarize their ideas in weekly previews and reports. Perhaps, in addition to this task, the instructors may ask the students to explain how newly learned knowledge can be applied to a different context (the apply level) or when the students are asked to check the weekly previews and reports of their peers, they may try to evaluate content and report their evaluation results, along with their feedback and ideas related to presentations (evaluate).

4.3.2. Homework

The results of the t-test to compare cognitive levels of homework in Study 1 and Study 2 are reported in Table 3. According to the results, the frequency of the remember level in Study 1 is significantly higher than that in Study 2 (t = 7.515; $p = 0.000 < 0.05$). However, the frequency of the understand level in Study 2 is significantly higher than that

in Study 1 (t = −2.244; $p$ = 0.046 < 0.05). Furthermore, the frequency of the create level in Study 2 is also significantly higher than that in Study 1 (t = −5.842; $p$ = 0.000 < 0.05).

**Table 3.** Cognitive levels assessment of homework of Study 1 and Study 2 and their comparison.

| Cognitive Levels | Group | Mean | SD | *t*-Value | *p*-Value |
|---|---|---|---|---|---|
| Remember | Study 1 | 7.25 | 2.121 | 7.515 | 0.000 * |
| | Study 2 | 0.00 | 0.000 | | |
| Understand | Study 1 | 213.88 | 90.802 | −2.244 | 0.046 * |
| | Study 2 | 406.40 | 218.759 | | |
| Apply | Study 1 | 214.63 | 102.802 | 1.503 | 0.161 |
| | Study 2 | 131.80 | 84.872 | | |
| Analyze | Study 1 | 21.75 | 13.562 | −2.028 | 0.110 |
| | Study 2 | 88.60 | 72.920 | | |
| Evaluate | Study 1 | 0.00 | 0.000 | | |
| | Study 2 | 0.00 | 0.000 | | |
| Create | Study 1 | 0.88 | 0.641 | −5.842 | 0.000 * |
| | Study 2 | 2.80 | 0.447 | | |

* $p$ < 0.05.

To explain these results, we need to refer to Figures 6 and 11. From the figures, we can find that cognitive level of the students rarely reaches low levels such as remember in Study 2. For this reason, the frequency of remember level in Study 1 is significantly higher than that in Study 2. However, the frequency of other levels of cognition such as understand and create, which are higher than remember, are higher in Study 2 compared to Study 1. Higher levels of cognition represent that students are more proactive in thinking and generating new ideas in Study 2 than Study 1. Interviews results from students after Study 2 can also support our findings (see Appendix A).

Our results show that intervention in Study 2 could facilitate the understand cognitive level. That is, students were able to understand their homework assignment better because of the hands-on practice in Study 2. The interview data also support this result (see Appendix A).

About the frequency of the create cognitive level, it was significantly higher in Study 2 than in Study 1. This result also proves that the intervention of Study 2 was indeed more useful to facilitate higher cognitive levels (i.e., create) than of Study 1. It also shows the success of Study 2 in teaching strategies and curriculum adjustment. Interview excerpts are provided in Appendix A.

### 4.3.3. Monthly Project Report

The results of the monthly project report evaluation are included in Table 4. In addition, it includes the results of the independent t-test, which was used to compare cognitive level of students in Study 1 and Study 2. According to the results, the frequency of evaluation levels in Study 2 was significantly higher than that in Study 1 (t = −5.000; $p$ = 0.038 < 0.05). From Figure 6 and Table 4, we can find that students did not reach the evaluate level in Study 1. In contrast, in Study 2, the students in each group could gradually judge and comment on different power quality events in the monthly project report according to the algorithm specifications, experience, and standards. This is the reason that explains the difference between two studies.

The data show that there was no significant difference in other cognitive levels between Study 1 and Study 2 (see Table 4). However, when referring to Figures 6 and 11, it can be found that the students had more instances of higher cognitive levels (i.e., analyze, evaluate, and create) in Study 2 than in Study 1. The results suggest that the course mode, the guidance of the teacher, and the strategies arrangement of Study 2 were obviously beneficial to promote the students' higher cognitive levels. Some objective evidence to support the results was derived from the interviews after Study 2 (see Appendix A).

**Table 4.** Cognitive levels assessment of monthly preview and reports of Study 1 and Study 2 and their comparison.

| Cognitive Levels | Group | Mean | SD | *t*-Value | *p*-Value |
|---|---|---|---|---|---|
| Remember | Study 1 | 2.2500 | 2.62996 | −1.063 | 0.337 |
| | Study 2 | 6.0000 | 6.55744 | | |
| Understand | Study 1 | 31.7500 | 12.09339 | −2.011 | 0.101 |
| | Study 2 | 53.3333 | 16.56301 | | |
| Apply | Study 1 | 30.5000 | 11.26943 | 0.151 | 0.886 |
| | Study 2 | 29.3333 | 8.14453 | | |
| Analyze | Study 1 | 21.2500 | 10.46821 | −1.627 | 0.236 |
| | Study 2 | 64.3333 | 44.97036 | | |
| Evaluate | Study 1 | 0.0000 | 0.00000 | −5.000 | 0.038 * |
| | Study 2 | 6.6667 | 2.30940 | | |
| Create | Study 1 | 1.0000 | 1.15470 | −1.435 | 0.211 |
| | Study 2 | 3.3333 | 3.05505 | | |

* $p < 0.05$.

### 4.3.4. Learning Outcomes

The students' learning outcomes in Study 1 and Study 2 are represented by the scores of their final projects. We used the independent t-test to compare learning outcome between Study 1 and Study 2. As shown in Table 5, the learning outcomes in Study 2 were significantly better than in Study 1 (t = −3.270, $p = 0.004 < 0.05$). This proves that the teaching strategies in Study 2 impacted student learning outcomes positively so that the student cognitive level improved. Some evidence was obtained from interviews with the students after Study 2 (see Appendix A).

**Table 5.** The independent t-test for the learning outcomes between Study 1 and Study 2.

| Study Number | Mean | SD | *t*-Value | *p*-Value |
|---|---|---|---|---|
| Study 1 | 88.83 | 1.749 | −3.270 | 0.004 * |
| Study 2 | 90.80 | 1.033 | | |

* $p < 0.05$.

In order to consider that the students' prior knowledge may have an impact on the learning outcomes, we also interviewed the instructor. The excerpts from the interview with the teacher are provided in Appendix A.

Based on the evidence from the interview, we can exclude the uncertain factors of the prior knowledge and ensure that learning outcomes were influenced by the teaching strategies, the guidance of the teacher, and the curriculum arrangement in Study 2.

The results demonstrate that various instructional strategies were beneficial for students in their learning. For example, in flipped classroom, students can prepare for their class at home and then spend class time on discussing new concepts they leaned [7]. Instructors then can easily identify student difficulties and misconceptions [8]. Project-based learning enabled students to identify the problem and then try to solve it in collaboration with other students [12,13]. Such an approach created authentic learning contexts in which problems students dealt with were those that they likely to experience in the real world [9,11]. The reciprocal teaching approach enabled students to become a teacher in a small group [16]. A student-teacher then guided group discussions using various strategies, e.g., summarizing, question generating, clarifying, and predicting [18]. All of these approaches were employed in the study to ensure student-centered learning, the interaction between the teacher and students, and the facilitation of cognitive levels. We found that flipped classroom and project-based learning had an impact on the cognitive-level development, especially on the understand and analyze levels. However, when reciprocal teaching was introduced, other cognitive levels improved as well, e.g., apply and analyze.

Based on our results, we suggest that diverse instructional approaches need to be implemented. Our results demonstrated that students did not have sufficient time in this course to master their professional knowledge and abilities. Therefore, we also suggest that the intervention last longer, e.g., one academic year. In this case, students will have long-term exposure to the treatment, and their cognitive abilities, such as various levels of cognition, can be developed diversely and become even better. From the results, we also found the importance of hands-on experience. Therefore, we suggest that students have hands-on experience (e.g., to perform and operate LabView) and sufficient time for practicing their skills so that their knowledge and skills can be validated and consolidated. We also suggest that the role of the teacher is important in the learning process when various instructional approaches are implemented (flipped classroom, project-based learning, and reciprocal teaching). The instructor can observe the learning process and interaction among students and intervene, when necessary, by assisting, guiding, and providing feedback to students when they experience any difficulties.

Some limitations regarding the present study need to be acknowledged. First, a small number of students participated in the study, and it was carried out over a limited period of time. Therefore, the findings of the present study need to be interpreted with caution. Another limitation is that all participants of the study were males, and this is because there are not so many female students who study electrical engineering. Future studies may consider increasing the number of their participants, carrying out their studies for a longer period of time, and considering involving female participants as well. Future studies may also consider exploring other research variables that may provide evidence for effective applications of reciprocal teaching in the flipped classroom to facilitate a high level of cognition for sustainable learning practices. For example, learning behavior and interaction among participants can be explored deeper, as well as employing path analysis to investigate common patterns in learning behavior and interaction during reciprocal teaching in the flipped classroom.

## 5. Conclusions

In the traditional classroom, students only listen to the teacher's lecture. We incorporated the flipped classroom strategy with weekly previews and reports to assist students to learn actively, PBL to help students have more ideas of learning by doing and strengthen their team works, and reciprocal teaching strategy to promote their collaboration and presentation skills. These approaches were applied to create a learner-centered environment and enhance students' cognitive skills toward professional knowledge from the basic-understanding cognitive level to higher cognitive levels, i.e., analyze, evaluate, and create. PBL and weekly previews and reports helped students preview what they learned and find and solve problems independently; hence, when the students achieved the learning goal, they felt satisfied. In addition, our approach was useful to improve their motivation to learn more.

In Study 2, we used student leadership and creativity abilities to group them heterogeneously. We found that it was helpful, as the students' higher levels of cognitive skills were improved, and it helped achieve sustainable learning practices. In the future, we may also consider other demographic factors, such as students' personality, background knowledge, expertise, etc.

Because students perceived that their learning load was heavy in Study 1, the teacher adjusted the class schedule promptly and reduced the homework amount to fit the curriculum to their needs and capacity in Study 2. Therefore, in Study 2, the students learned based on their paths. Our results showed that the students were pleased and satisfied with this course.

**Author Contributions:** Conceptualization, methodology, formal analysis, investigation, resources, data curation, writing—original draft preparation, W.-Y.H. and T.-H.W.; validation, writing—review and editing, correspondence, R.S.; supervision, funding acquisition, W.-Y.H. All authors have read and agreed to the published version of the manuscript.

**Funding:** This research was partly funded by 109-2511-H-008-009-MY3 and 111-2410-H-008-061-MY3, National Science and Technology Council, Taiwan.

**Institutional Review Board Statement:** The study was conducted in accordance with the Declaration of Helsinki and approved by the Institutional Review Board of Graduate Institute of Network Learning Technology at the National Central University for studies involving humans.

**Informed Consent Statement:** Informed consent was obtained from all subjects involved in the study.

**Data Availability Statement:** The data presented in this study are available upon request from the corresponding author. The data are not publicly available due to restrictions such as privacy and ethical reasons.

**Conflicts of Interest:** The authors declare no conflict of interest.

## Appendix A

Extracts from interviews with students:

About Homework

*"Every week we have homework, I feel it is too burdened and time is tight, but I still find the course design very helpful."*

*"It takes a considerable amount of time from the understanding of the algorithm to the actual test."*

*"The implementation software may require additional teaching and assistance, plus the preparation for flipped reports and project reports, which often causes the progress of the work not easy to keep up."*

About Study 1

*"Weekly preview and report helped us better understand the content of the course, and we could also know the advantages and disadvantages of various methods and applications in advance. The most important thing was that I think the implementation made me more conceptual about the content of the course."*

*"I suggest that if we can take this course in the computer classroom, it can help us improve the application skills and project completion of LabView software."*

*"These teaching strategies have prompted us to complete the preview. Students need to find information to understand it, we can't just blindly memorize the formula and don't know how to apply it. I am very fulfilled in this course. (I feel that the all practical classes will also give us a sense of accomplishment, and the different part of this class is not just to follow the experimental steps but to know what we are doing)."*

About Weekly Preview and Report

*"The learning ability has been greatly improved and it can be applied continuously."*

*"I have learned how to interact and collaborate with group members."*

*"No matter knowledge or expression ability, I have a big harvest."*

About Reciprocal Teaching Activity

*"After group discussions, we can better understand the curriculum and enhance our ability to express."*

*"There is more opportunity and time to ask questions and discuss."*

About Hands-On Practice using LabView

*"The hands-on practice is like experimenting in general. If we encounter problems, we could think about how to solve them by ourselves. If we don't know how to solve them, we just raise our hand to ask the teacher."*

About Monthly Project Report

*"Increasing opportunities to communicate with colleagues."*

*"We could improve each other's content during the discussion."*

About Homework

*"This learning model is easier for us to understand what we have learned and how it is applied."*

*"After understanding the theory, we will improve the areas where the theory is insufficient and increase our own ideas."*

*"It can help us to understand the content of the class easier."*

*"When we finish our homework, we will feel fulfilled."*

*"Through the hands-on practice, we can understand the theory of the application more clearly. And when we finish the implementation of LabView, we will feel fulfilled."*

*"Sometimes when I do some homework, I will recall I might have ever learned it in class before, and I just know what the teacher taught us at that time, so it will make me extend different ideas."*

*"I have other insights and innovative ideas for the application of the test methods, and they are applied to the project."*

*"In this way of learning, it let me know how to apply various methods."*

*"Under this learning mechanism, every time is to break through myself."*

About Monthly Project Report

*"The teacher explained the course content clearly and he answered questions I don't understand."*

*"I like this way of class. I will preview the course content before class. In this way, I can know the content first before the teacher teach us and help myself to understand the content better. And I will not feel so hard to understand when we start class, and the teacher will give us time to practice in the classroom and allow us to have time to discuss with each other, so that I can clearly understand the theory of the implementation, and finally we can share our ideas between groups. The new ideas and methods are very helpful."*

*"It let me learn a lot of things from the preview discussion and the review discussion."*

About Learning Outcomes

*"I want to score this class more than 100 points. The teacher discussed with the students in addition to teaching, so that we could more easily absorb the content of the class."*

*"I think this class was very good and want to recommend everyone to study. I have learned a lot in this course and it was very helpful to me. Thanks the teacher for teaching and hard work this semester, thank you so much!"*

*"This class had more hands-on practice than the general theoretical course. Fortunately, the teacher adjusted the teaching strategies of the course, otherwise the load was very heavy in class; because I wanted to learn the theoretical basis originally, and I did not expect that we will have hands-on practice. The ability of the teacher and classmates made me want to strengthen my ability. Although the things I usually study are less relevant to this course, I tried to integrate what I have learned in this course into my own research in the future."*

*"This course is for the master degree students. The students have learned about engineering mathematics, signal and system courses at the university, so they have the prior knowledge, and if they have not learned about relevant course in this fields from their"*

*universities, and they do not need to worry about it, because when we explain the relevant knowledge and skills in the classroom, and these things are introduced from the basics, it is not difficult to get started, and it is not affected by the prior knowledge."*

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
