# Peer review of "Applications of Reciprocal Teaching in Flipped Classroom to Facilitate High Level of Cognition for Sustainable Learning Practices"

_sustainability, doi:10.3390/su15075848_

Round 1
Reviewer 1 Report
The authors aimed to promote students’ skills and enhance their high level cognitive abilities by incorporating reciprocal teaching method in flipped classroom into project-based engineering curriculum and achieve sustainable teaching and learning practices. The paper seems interesting but requires major improvement.
1- The literature review should be extended. Most of the cited studies are relevantly old. Please add the recent studies.
2- The contribution of the paper is unclear to me. This comes from my previous comment. As the literature is not discussed in detail, the gaps are vague.
3- The discussion section should be revised. The applications of the current study, limitations, and future research are missing.
4- the visualization must be improved. For example, figure 4 is incomplete. The other figures also require a better presentation.
5- The language needs minor improvement as well.
Author Response
Comment 1.1:
The authors aimed to promote students’ skills and enhance their high level cognitive abilities by incorporating reciprocal teaching method in flipped classroom into project-based engineering curriculum and achieve sustainable teaching and learning practices. The paper seems interesting but requires major improvement.
1- The literature review should be extended. Most of the cited studies are relevantly old. Please add the recent studies.
Response: 1.1:
Dear Reviewer 1, thank you for your valuable comments and suggestions. They were useful for improving the quality of our submission.
For the first comment, we have added more recent studies to the paper:
- García-Segura, T., Montalbán-Domingo, L., Sanz-Benlloch, A., Domingo, A., Catalá, J., & Pellicer, E. (2023). Enhancing a Comprehensive View of the Infrastructure Life Cycle through Project-Based Learning. Journal of Civil Engineering Educa-tion, 149(1), 05022002.
- Jia, C., Hew, K. F., Jiahui, D., & Liuyufeng, L. (2023). Towards a fully online flipped classroom model to support student learning outcomes and engagement: A 2-year design-based study. The Internet and Higher Education, 56, 100878.
- Jiang, C., & Pang, Y. (2023). Enhancing design thinking in engineering students with project‐based learning. Computer Applications in Engineering Education, 1-17.
- Merrett, C. G. (2023). Analysis of Flipped Classroom Techniques and Case Study Based Learning in an Introduction to En-gineering Materials Course. Advances in Engineering Education, 11(1), 2-29.
- Zewail-Foote, M., & Gonzalez, M. (2023). Crisscrossing Learning Experiences in an Undergraduate Research-Based Labor-atory Course to Promote Reciprocal Peer Learning. Journal of Chemical Education.
- Zhang, H., Zhang, J., Wang, J., & Zhang, H. (2023). Integrating mobile learning and SPOC‐based flipped classroom to teach a course in water supply and drainage science and engineering. Computer Applications in Engineering Education, 1-14.
Comment 1.2:
2- The contribution of the paper is unclear to me. This comes from my previous comment. As the literature is not discussed in detail, the gaps are vague.
Response: 1.2:
We have included the following paragraph to show our contribution:
Informed by related studies, we applied reciprocal teaching in flipped classroom to facilitate high level of cognition for sustainable learning practices. The learning ac-tivity was designed following project-based learning (PBL) methodology. With such integrative approach, we aimed to cultivate and develop engineering students’ high level cognitive abilities. As can be seen from the literature review, not many studies had such integrative approach or focused on engineering students’ high level cognitive abilities development.
Comment 1.3:
3- The discussion section should be revised. The applications of the current study, limitations, and future research are missing.
Response: 1.3:
We have included missing details about the applications of the current study, limitations, and future research in the paper:
Some limitations regarding the present study need to be acknowledged. First, small number of students participated in the study and it was carried out over limited period of time. Therefore, the findings of the present study need to be interpreted with caution. Future studies may consider increasing the number of their participants and carry their studies out for longer period of time. Future studies may also consider exploring other research variables that may evidence for effective applications of reciprocal teaching in flipped classroom to facilitate high level of cognition for sustainable learning practices. For example, learning behavior and interaction among participants can be explored deeper as well as employing path analysis to investigate common pat-tern in learning behavior and interaction during reciprocal teaching in flipped classroom.
Comment 1.4:
4- the visualization must be improved. For example, figure 4 is incomplete. The other figures also require a better presentation.
Response: 1.4:
We have improved the visualization of our figures and Figure 4 was presented in full.
Comment 1.5:
5- The language needs minor improvement as well.
Response: 1.5:
We have improved English writing in the paper.
Reviewer 2 Report
Dear Authors,
I have reviewed your article titled "Incorporating Reciprocal Teaching Method into Flipped Classroom for Sustainable Learning Practices" and I must say that it presents interesting findings on the use of the reciprocal teaching method to improve sustainable learning practices in project-based engineering education. Overall, the article is well written and organized, and it contributes to the field of education.
However, I would like to request that some revisions be made to the manuscript before it can be considered for publication. Firstly, I noticed that Figure 4 needs to be reworked, and the spaces between the lines of the tables in the appendix need to be adjusted. Additionally, I recommend that the appendices be included within the main text of the article, as they do not need to be separate from the rest of the manuscript. With these minor formatting improvements, I believe your work can be approved for publication.
Thank you for your contributions to the field, and I look forward to seeing the revised version of your manuscript.
Best regards,
Reviewer
Author Response
Comment 2.1:
Dear Authors,
I have reviewed your article titled "Incorporating Reciprocal Teaching Method into Flipped Classroom for Sustainable Learning Practices" and I must say that it presents interesting findings on the use of the reciprocal teaching method to improve sustainable learning practices in project-based engineering education. Overall, the article is well written and organized, and it contributes to the field of education.
However, I would like to request that some revisions be made to the manuscript before it can be considered for publication. Firstly, I noticed that Figure 4 needs to be reworked, and the spaces between the lines of the tables in the appendix need to be adjusted. Additionally, I recommend that the appendices be included within the main text of the article, as they do not need to be separate from the rest of the manuscript. With these minor formatting improvements, I believe your work can be approved for publication.
Thank you for your contributions to the field, and I look forward to seeing the revised version of your manuscript.
Best regards,
Reviewer
Response: 2.1:
Dear Reviewer 2, thank you for your valuable comments and suggestions. They helped us to improve the quality of our manuscript a lot. Following your suggestion, we reworked Figure 4 so it is clearer now. We have also included the following description regarding the figure:
The data and content presented by the students were classified into six levels of the taxonomy with different colors (Figure 4). For example, green color represents remember level and yellow color represents create level (see Figure 3 for colors and related levels of the taxonomy).
In addition, as suggested, we have relocated tables from Appendix B to their corresponding locations in the main text.
Reviewer 3 Report
The abstract is informative and seems to address all necessary information.
The literature review is brief and dated; only two citations from 2020. Flipped Classroom and PBL are rigorously researched areas, and require more current articles. The citation format does not align with the reference format; requires attention.
Collaborative teaching (Study 1) and Reciprocal Teaching (Study 2) are very similar in practice.; because it aligns with Vygotsky's socio construtivist theory.
The Methodology needs tidying up. The reference includes a text about Case Study, yet the methodology does not explain the method for conducting this research. Case Study needs to be more clearly stated.
Study 1 & 2 explain the number of participants, and does not clarify the gender ratio, the age, and the year level (Junior = ?) / freshman? sophomore?
The article is primarily an opinion essay reporting on the findings.
Figures 4 & 5 are in Chinese. This needs to be improved; the relevance of these images cannot be determined.
There is no reporting on the teaching involvement. For example, aligning figure 13 with teaching practice that led to the increase of analysis skills.
The article seems to be an unfinished paper which is not yet ready for publication.
Zulkifli, N., Abd Halim, N., Yahaya, N., Meijden, H., Zaid, N. M., Rashid, A. A., & Hashim, S. (2021). Online reciprocal peer tutoring approach in Facebook: Measuring students' critical thinking. International Journal of Emerging Technologies in Learning (iJET), 16(23), 16-28.
Author Response
Comment 3.1:
The abstract is informative and seems to address all necessary information.
The literature review is brief and dated; only two citations from 2020. Flipped Classroom and PBL are rigorously researched areas, and require more current articles. The citation format does not align with the reference format; requires attention.
Response: 3.1:
Dear Reviewer 3, thank you for your valuable comments and suggestions. We tried out best to address them all. Please find our detailed responses below.
To address the first comment, we have added more recent references:
- García-Segura, T., Montalbán-Domingo, L., Sanz-Benlloch, A., Domingo, A., Catalá, J., & Pellicer, E. (2023). Enhancing a Comprehensive View of the Infrastructure Life Cycle through Project-Based Learning. Journal of Civil Engineering Educa-tion, 149(1), 05022002.
- Jia, C., Hew, K. F., Jiahui, D., & Liuyufeng, L. (2023). Towards a fully online flipped classroom model to support student learning outcomes and engagement: A 2-year design-based study. The Internet and Higher Education, 56, 100878.
- Jiang, C., & Pang, Y. (2023). Enhancing design thinking in engineering students with project‐based learning. Computer Applications in Engineering Education, 1-17.
- Merrett, C. G. (2023). Analysis of Flipped Classroom Techniques and Case Study Based Learning in an Introduction to En-gineering Materials Course. Advances in Engineering Education, 11(1), 2-29.
- Zewail-Foote, M., & Gonzalez, M. (2023). Crisscrossing Learning Experiences in an Undergraduate Research-Based Labor-atory Course to Promote Reciprocal Peer Learning. Journal of Chemical Education.
- Zhang, H., Zhang, J., Wang, J., & Zhang, H. (2023). Integrating mobile learning and SPOC‐based flipped classroom to teach a course in water supply and drainage science and engineering. Computer Applications in Engineering Education, 1-14.
About the citation format, we know about this problem in our paper and we worry that there will be more requests to add some new references in the next rounds of review. Because this journal follows very specific referencing and citation format, if we add new references then the whole reference list and citations need to be changed. We will correct the citation format in the paper after there are no comments from the reviewers. Thank you for your understanding.
Comment 3.2:
Collaborative teaching (Study 1) and Reciprocal Teaching (Study 2) are very similar in practice.; because it aligns with Vygotsky's socio construtivist theory.
Response: 3.2:
Thank you for this comment, we have emphasized this point in the paper:
Both collaborative learning (Study 1) and reciprocal teaching (Study 2) align with social constructivism theory (Vygotsky, 1978). According to this theory, learners are active participants in the creation of their own knowledge. It was suggested that a central notion in social constructivism is assisted learning, a concept that is influenced by socio-culturalism and its concept of proximal learning. Although both activities align with social constructivism theory, they are quite different in nature and as we assumed brings different learning effects.
Comment 3.3:
The Methodology needs tidying up. The reference includes a text about Case Study, yet the methodology does not explain the method for conducting this research. Case Study needs to be more clearly stated.
Response: 3.3:
We have added more details to explain our research methodology in the paper:
The research method was a case study (Eisenhardt, 1989). According to Creswell (2012), a case study is an in-depth exploration of a bounded system (e.g., activity, event, or process) based on extensive data collection. That is, in a case study, researchers focus on a program, event, or activity involving individuals or a group. We particularly, employed multiple instrumental case study. We focused on illuminating a specific issue (how to facilitate high level of cognition for sustainable learning practices) with cases (Study 1 and Study 2) used to illustrate the issue. We described and compared cases to provide insight into an issue (Herreid & Schiller, 2013)
Comment 3.4:
Study 1 & 2 explain the number of participants, and does not clarify the gender ratio, the age, and the year level (Junior = ?) / freshman? sophomore?
Response: 3.4:
We have added the missing information:
In Study 1, twelve junior graduate students majoring in electrical engineering participated. All of them were males.
In Study 2, ten junior graduate students with the major in electrical engineering participated. All of them were males.
Comment 3.5:
The article is primarily an opinion essay reporting on the findings.
Response: 3.5:
Thank you.
Comment 3.6:
Figures 4 & 5 are in Chinese. This needs to be improved; the relevance of these images cannot be determined.
Response: 3.6:
Thank you.
Comment 3.7:
There is no reporting on the teaching involvement. For example, aligning figure 13 with teaching practice that led to the increase of analysis skills.
Response: 3.7:
We mentioned about the role of the teacher and teaching assistant in this study:
The teacher and teaching assistant helped the students when necessary. For example, the students asked a teacher or teaching assistant questions when they encountered any difficulties.
Comment 3.8:
The article seems to be an unfinished paper which is not yet ready for publication.
Zulkifli, N., Abd Halim, N., Yahaya, N., Meijden, H., Zaid, N. M., Rashid, A. A., & Hashim, S. (2021). Online reciprocal peer tutoring approach in Facebook: Measuring students' critical thinking. International Journal of Emerging Technologies in Learning (iJET), 16(23), 16-28.
Response: 3.8:
Thank you. We tried our best to improve the quality of this study.
Round 2
Reviewer 1 Report
Thank you for considering my comments and addressing them. I have no further comments on your work.
Author Response
Thank you!
Reviewer 3 Report
Page 7 lines 278~279 could be improved.
Figure 5 remains in Chinese and needs to be update for the English Speaking community. Otherwise remove as it is considered exclusive and closed to Non Chinese speakers.
The authors may want to explain further in the discussion, the strategies the literature recommends for improving Items in Table 2. How can teachers and students improve the concept "create", this may be informative for the audience, and the authors should they decide to continue the research.
Author Response
Dear Reviewer, thank you for your comments!
"Page 7 lines 278~279 could be improved." - we have revised our statements.
"Figure 5 remains in Chinese and needs to be update for the English Speaking community. Otherwise remove as it is considered exclusive and closed to Non Chinese speakers." - we have removed the figure from the paper.
"The authors may want to explain further in the discussion, the strategies the literature recommends for improving Items in Table 2. How can teachers and students improve the concept "create", this may be informative for the audience, and the authors should they decide to continue the research." - we have added our suggestions regarding how to improve higher cognitive levels of the students, please see the end of the "4.3.1. Weekly Preview and Report" section.